# Protocol for the first large-scale emergency care-based longitudinal cohort study of recovery after sexual assault: the Women's Health Study

Nicole A Short,[1,2] Jenyth Sullivan,[2] April Soward,[2] Kenneth A Bollen,[3] Israel Liberzon,[4] Sandra Martin,[2] Sheila A M Rauch,[5] Kathy Bell,[6] Catherine Rossi,[7] Megan Lechner,[8] Carissa Novak,[2] Kristen Witkemper,[2] Ronald C Kessler,[9] Samuel A McLean[2,10]

For numbered affiliations see end of article.

**Correspondence to**
Dr Samuel A McLean;
Samuel_McLean@med.unc.edu

## ABSTRACT

**Introduction** Worldwide, an estimated 10%–27% of women are sexually assaulted during their lifetime. Despite the enormity of sexual assault as a public health problem, to our knowledge, no large-scale prospective studies of experiences and recovery over time among women presenting for emergency care after sexual assault have been performed.

**Methods and analysis** Women ≥18 years of age who present for emergency care within 72 hours of sexual assault to a network of treatment centres across the USA are approached for study participation. Blood DNA and RNA samples and brief questionnaire and medical record data are obtained from women providing initial consent. Full consent is obtained at initial 1 week follow-up to analyse blood sample data and to perform assessments at 1 week, 6 weeks, 6 months and 1 year. These assessments include evaluation of survivor life history, current health and recovery and experiences with treatment providers, law enforcement and the legal system.

**Ethics and dissemination** This study is approved by the University of North Carolina at Chapel Hill's Institutional Review Board (IRB) and the IRB of each participating study site. We hope to present the results of this study to the scientific community at conferences and in peer-reviewed journals.

## INTRODUCTION

Worldwide, an estimated 10%–27% of women are sexually assaulted during their lifetime.[1] In the USA, as many as 1.9 million women are sexually assaulted each year.[2 3] Survivors of sexual assault experience a heavy burden of adverse post-traumatic neuropsychiatric sequelae (APNS), including fear, anxiety and post-traumatic stress.[4] In addition, as many as half of sexual assault survivors experience clinically significant depression,[5] one quarter report suicidal ideation and 2%–19% attempt suicide.[4] Preliminary evidence from small prospective cohorts also suggests that acute and chronic pain and somatic symptoms

## Strengths and limitations of this study

► To our knowledge, this is the first large-scale prospective study of experiences and recovery among women presenting for emergency care after sexual assault.

► The study obtains detailed longitudinal data from the sexual assault survivors on adverse post-traumatic neuropsychiatric sequelae (APNS) over time after sexual assault, including post-traumatic stress, pain, somatic, cognitive and depressive symptoms.

► The study collects both biologic data to gain new insights into the pathogenesis of APNS after sexual assault, and data to develop clinical prediction tools that identify those at high risk of APNS.

► The study obtains feedback from women sexual assault survivors regarding their experience with police, the legal system and medical providers that can be used to improve services.

► Generalisability of study findings to women sexual assault survivors who do not report/come for care after sexual assault is not known.

frequently develop after sexual assault.[6–8] Amid this tremendous physical and mental health suffering, 13%–49% of sexual assault survivors develop an alcohol use disorder and 28%–61% develop problematic use of other substances.[9 10]

In the USA, emergency care for women sexual assault survivors is most commonly provided by a nurse with specific training and expertise, termed a Sexual Assault Nurse Examiner (SANE). The emergency care of sexual assault survivors provided by a SANE includes a thorough medical and forensic history, a detailed physical examination to document evidence of trauma, evidence collection and risk stratification and preventive interventions to reduce pregnancy and/or sexually transmitted disease.

Unfortunately, no risk stratification and preventive interventions are employed to reduce APNS. This is because risk stratification tools for APNS do not exist, and because the developmental biology of ANPS is poorly understood.

This article describes the study protocol for the first large-scale emergency care-based longitudinal cohort study of recovery after sexual assault: the Women's Health Study (WHS)

. The goals of the study are to help address the above barriers to intervention, gain a better understanding of survivor experiences with treatment providers, law enforcement and legal and healthcare systems to inform improvement, and to determine the incidence and recovery trajectories of a range of APNS.

## METHODS/DESIGN
### Study sites
The WHS is a prospective, multisite observational study of women presenting for emergency SANE care following sexual assault. Women are recruited at emergency care centres across the USA that are part of the Better Tomorrow Network, including Albuquerque SANE Collaborative, Albuquerque, New Mexico; Tulsa Forensic Nurse Examiners, Tulsa, Oklahoma; Christiana Care SANE Program, Newark, Delaware; Austin Stop Abuse for Everyone (SAFE), Austin, Texas; Hennepin Hennepin Assault Response Team (HART), Minneapolis, Minnesota; Crisis Center of Birmingham, Birmingham, Alabama; Philadelphia Sexual Assault Response Center (SARC)/Einstein Healthcare, Philadelphia, Pennsylvania; Denver Health SANE, Denver, Colorado; Wayne County SAFE, Detroit, Michigan; University of Louisville SAFE, Louisville, Kentucky; Memorial Health SANE, Colorado Springs, Colorado; DC SANE, Washington, DC; and Cone Health, Greensboro, North Carolina. Each site's institutional review board (IRB) approved the study protocol. The Better Tomorrow Network data coordinating centre is located at the University of North Carolina (UNC) at Chapel Hill North Carolina, USA.

### Patient inclusion criteria
Women at least 18 years of age who present to a participating site for emergency care within 72 hours of assault are assessed for potential eligibility. Exclusion criteria include: inability to provide informed consent (eg, due to intoxication, injury), pregnancy, living with one's assailant, presence of fracture, injury requiring hospital admission, inability to speak or read English, no telephone access, no mailing address, unwilling to provide a blood sample, incarceration and other situations resulting in inability to follow the study protocols. Enrolment began in June 2015, study follow-ups are scheduled to be completed in 2020, with a target enrolment of 700 women.

### Patient screening and initial assessment
Treatment providers at study network sites notify on-call research associates (RAs) when a potentially eligible individual presents for emergency care after sexual assault. RAs approach potential participants at a time determined by the sexual assault survivor care provider(s). RAs provide a brief description of the study and request written consent to (1) perform a brief survey, (2) contact the survivor by telephone in 48–72 hours to explain the full study and ask if the individual is interested in participating, (3) access medical record/forensic examination information related to the sexual assault and (4) collect blood samples (discarded if the participant does not subsequently consent to the full study at 1 week). Individuals are specifically informed that they are not being asked to decide about full study participation at the time of their emergency care. Participants are compensated $20 for completing the initial assessment.

RAs communicate with potential participants regarding consent for the initial evaluation either in-person or via the use of teleconsent. To obtain teleconsent, the RA introduces her/himself to the participant via live two-way video communication over tablet or laptop, and describes her/his role as an RA in an ongoing study to learn more about the recovery process after trauma. Just as in-person, if the potential participant is willing, the RA reviews the study consent and requirements for participation and uses the online system to screen the potential participant for study eligibility. If eligible, the RA opens a copy of the initial consent on the tablet or laptop, and obtains an e-signature. The teleconsent platform complies with Health Insurance Portability and Accountability Act requirements by (1) encrypting all transmitted data, (2) not storing patient information and (3) keeping an audit trail of the consent process.[11] This teleconsent is used as a consent option because it has been shown to be a safe and valid method of obtaining consent in medical settings[11–18] and because it addresses barriers that prevent critically needed research in settings where it is difficult to consistently provide trained research staff. Teleconsent also addresses inequities created by the fact that research staff are more difficult to hire/provide in socioeconomically disadvantaged and/or rural areas.

### Assessment at 1 week
Participants who are reached by phone and express interest in study participation are scheduled for a 1 week in-person assessment. Assessments take place either at the initial emergency care site or at a public community location acceptable to both the RA and study participant. At the beginning of the study visit, the RA obtains written informed consent to participate in follow-up evaluations 1 week, 6 weeks, 6 months and 1 year after the assault and consent to utilise the blood samples obtained at the time of emergency care. After written informed consent is obtained, the 1 week evaluation is typically performed via a web-based self-report computer survey completed on a laptop computer. Use of a computerised self-report survey standardises the assessment experience across RAs and sites, increases reporting of experiences that may be considered stigmatising,[19] and facilitates honest

feedback regarding the experience of being in the study. Computer self-report surveys have been shown to be a valid and acceptable means of obtaining research assessment data, including in socioeconomically disadvantaged populations recruited in emergency care settings.[20–22] The beginning of each web-based assessment includes introductory practice questions using the different questionnaire formats. After the completion of these practice questions, the RA moves to another location in the room, so that the study participant has privacy when recording her computer questionnaire responses. Participants are compensated $50 for completing the 1 week assessment.

### Assessment at 6 weeks, 6 months and 12 months

Follow-up survey assessments at 6 weeks, 6 months and 12 months may be completed in-person, or if the participant prefers, via logging into a secure website on a computer of their choice, or over the phone. The format of the web-based computer self-report surveys completed online are identical to the web-based computer self-report surveys completed by participants who elect to complete surveys in-person. Each online assessment contains specific completion instructions. Participants who complete follow-up assessments online independently may call the toll-free study phone number if they have any questions while completing the assessment. Participants are encouraged to complete these follow-up assessments either in-person or online, to maximise standardisation. However, if participants are unable to complete these follow-up assessments either in-person or online, they may also complete them via telephone interview. Participants are compensated $50 for completing each follow-up assessment.

### Contact information updates at 3 months and 9 months

Participants are contacted 3 months and 9 months after enrolment, and are asked to update their contact information. Participants are compensated $10 for each follow-up request.

### Confidentiality and security of participant data

All paper forms are stored in locked file cabinets with specially made keys at the Better Tomorrow Network Data Coordinating Center in Chapel Hill, North Carolina. These forms are labelled with participant ID only. Consent forms are stored separately because they contain personally identifying information. Study data are housed on a secure, fire-walled server dedicated for research use only and maintained by the UNC School of Medicine. Data are stored on an SQL Server, with encryption and full daily and nightly backups. Both the Application server and the Database server are located behind The UNC School of Medicine firewall, and only approved site users have access via ASP.NET Forms Authentication or Windows Integrated Authentication.

### Patient and public involvement

The study protocol was designed in collaboration with SANEs; this team included sexual assault survivors. The study was reviewed and funded by the National Institutes of Health, which considers public funding priorities and maximising the public good in scientific review and funding decisions. As described below, study participants provide feedback on burden and experiences in the research study at the end of each assessment. Study results are disseminated to the public via presentations, publications and media coverage, and via provision of study results to individual sexual assault treatment facilities and to state coalitions against sexual assault in states in which participating care sites are located.

### Study assessments

#### Assessments and data collection at the time of emergency care

Pain severity (0–10 numeric rating scale score; table 1) is assessed in each of 23 body regions using an adapted version of the Regional Pain Scale.[23] This scale has been shown to be a valid, reliable measure of pain location and distribution.[23] Pain severity in the head and face, breast area, pelvic/genital region, hands and feet are also assessed. Blood DNA and RNA are collected using PAXgene DNA (8.5cc) and RNA (2.5cc) storage tubes, respectively. Samples are frozen at the study site and are shipped in batches to the UNC. A barcode-based study tracking system tracks each tube throughout the storage and shipment process to maintain chain of custody.

Medical records regarding emergency care after sexual assault are password protected and then uploaded to the encrypted, secure study database by the site RA. If some of the participant's emergency care was provided at a treatment location separate from the location of the SANE examination, these records are also obtained by study staff. Medical record data are abstracted by RAs using detailed protocols and variable definitions. Data extracted include detailed information regarding the assault, medical care received, medical history and assailant relationship with the survivor (eg, relatives, current or former romantic partners, friend/acquaintance, planned first encounter, stranger or unknown.) Quality assurance and control of data abstraction are assessed via double data entry and discrepancies are adjudicated by the study Principal Investigator (PI).

#### Assessments at 1 week, 6 weeks, 6 months and 1 year

Assessments at 1 week, 6 weeks, 6 months and 1 year include the following:

##### General physical and mental health status

The Patient-Reported Outcomes Measurement Information System (PROMIS) Global Health assessment is used to assess general preassault health at the 1-week follow-up. At subsequent time points, this measure is used to assess health status.[24]

##### Somatic symptoms

Preinjury somatic symptoms during the week prior to the sexual assault are assessed at 1-week follow-up using a 21-question symptom inventory. Somatic symptoms

**Table 1** Study question domains and self-report measures administered at each time point

| Domain | Measure | Initial | 1 Wk | 6 Wk | 6 Mo | 12 Mo |
|---|---|---|---|---|---|---|
| Assault characteristics | Data extraction from medical record | X | | | | |
| Acute pain | Pain Severity Numeric Rating Scale[58] | X | X | X | X | X |
| Pain related to the assault | Pain Severity Numeric Rating Scale[58] | | X | X | X | X |
| Pain interference | Brief Pain Inventory[59] | | X | X | X | X |
| Preassault pain | Pain Severity Numeric Rating Scale[58] | | X | | | |
| Current medication use | Standard Items | | X | X | X | X |
| Survivor experience | SANE Care, advocate, postexposure prophylaxis | | X | | | |
| | Experience with police, postexposure prophylaxis, health-services utilisation, out-of-pocket expenses | | | X | | |
| | Experience with police, health services utilisation, out-of-pocket expenses | | | | X | |
| | Experience with police, experience with legal system, health services utilisation, out-of-pocket expenses | | | | | X |
| Somatic symptoms | Numeric rating scale score of 21 common symptoms | | X | X | X | X |
| Preassault somatic symptoms | Standard Items | | X | | | |
| Post-traumatic stress symptoms | PTSD Checklist–Situation (PCL-S)[26][27] | | X | X | X | X |
| Preassault trauma exposure | Adapted version of the Life Events Checklist[25] | | X | | | |
| Preassault post-traumatic stress | Adapted version of PCL-5[26][27] | | X | | | |
| Childhood trauma exposure | Childhood Adversity Exposures (ACE)[60] | | | X | | |
| Resilience | Trait Resiliency[28] | | X | | | |
| General health | PROMIS Global Health – Physical Component[24] | | X | X | X | X |
| New health status | Standard items | | | X | X | X |
| Depression | PROMIS – Short Form 8b[24] | | X | X | X | X |
| Anxiety | PROMIS – Short Form 8a[24] | | X | X | X | X |
| Substance use | Adapted version of CIDI-SC[29] | | X | X | X | X |
| Demographics | Standard items | | X | | | |
| Research experience | Reactions to Research Participation Questionnaire Revised[38] | | X | X | X | X |

CIDI-SC, Composite International Diagnostic Interview–Screening Scale; Mo, month; PROMIS, Patient-reported Outcomes Measurement Information System; PTSD, post-traumatic stress disorder; SANE, sexual assault nurse examiner; Wk, week.

during the past week are additionally assessed at each follow-up time point.

### Post-traumatic stress disorder symptoms, past trauma and resilience

Lifetime sexual assault and other trauma exposure and post-traumatic stress disorder (PTSD) history are assessed at 1 week using questions from an adapted version of the Life Events Checklist.[25] Lifetime PTSD symptoms related to prior trauma are assessed with an abbreviated and adapted version of the PTSD Checklist (PCL) 5[26 27] that explicitly instructs the respondent to focus only on symptoms related to prior stressors, not the recent sexual assault. An adapted version of the Trait Resilience scale of Kessler et al[28–30] is administered at 1 week and the Adverse Childhood Experiences[31 32] Score Calculator is administered at 6 weeks to assess whether women experienced childhood sexual assault or other forms of trauma.[31] Recent sexual assault-related PTSD symptoms are assessed using the PCL-5[27 32] with the addition of the two Diagnostic and Statistical Manual - 4th Edition (DSM-IV)-related items from the PTSD Checklist - Civilian version (PCL-C).[26 27 33]

### Depressive and other anxiety-related assessments symptoms

Short-form versions of the PROMIS Depression and Anxiety assessments[24] (Short Form 8b and Short Form 8a, respectively) are used at the 1-week follow-up to assess depressive and anxiety symptom burden in the week prior to assault. These assessments are also used at the 6-week, 6-month and 12-month follow-ups to assess symptoms during the past week. In addition, a brief version of the Anxiety Sensitivity Index-3 is administered at 1-week follow-up.[34 35]

### Pain symptoms

The same assessment method used to evaluate the severity and location of pain at the time of SANE examination are used to evaluate both reported preassault pain and postassault pain symptoms. Pain symptoms assessed at each time point are specifically termed 'pain/tenderness' and 'pain or aching', as past experience with sexual assault survivors indicates that some survivors otherwise interpret questions about 'pain' to include 'emotional pain'. At the 1-week assessment, after being asked about the location and severity of any pain symptoms since the assault, participants are also asked about the location and severity of any pain symptoms during the week before assault. If participants report pain in a body region at 6-week, 6-month or 12-month follow-up, after rating the severity of pain in that region participants are asked to indicate if this pain is related to the sexual assault. In addition to pain severity assessments in each body region, at each follow-up timepoint overall pain is also assessed using a 0–10 Numeric Rating Scale (NRS) and pain interference during the past week is assessed using questions from the Brief Pain Inventory.[36]

### Substance use

Drug, alcohol and tobacco use during the month prior to the month of assault are assessed at the 1-week follow-up using an adapted version of the Composite International Diagnostic Interview Screening Scale assessment of substance use disorder of Kessler et al.[29 30] This time point was chosen to exclude the timeframe of the assault. At all subsequent follow-ups, use during the past month is assessed.

### Demographic information

Information including age, race/ethnicity, education level, income level, marital status, work status, and height and weight are obtained from questionnaire items and the medical record.

### Medical history

Participant medical history is obtained from the medical records.

### Medication use

Participant medication use is assessed at each follow-up evaluation.

### Experiences of sexual assault survivors with care providers and the healthcare and legal systems

Sexual assault survivor experiences, including experiences with the SANE programme, legal system and police, as well as health services utilisation and injuries since assault, are assessed using standardised questionnaires. Legal system assessments include an evaluation of experiences with reporting the assault to police, interactions with prosecutors, whether there was an assault investigation was performed (and if not performed, reason(s) the survivor was told there was no investigation), and status/outcomes of the legal process.

### Sexual assault survivor experiences with the study in general and with individual follow-up assessments

After each interview, participants complete an adapted version of the Reactions to Research Participation Questionnaire[37 38] and open-ended questions about their experiences as a study participant and sexual assault survivor. Open-ended questions ask survivors the following: 'Is there anything about this survey that we could do better?', and 'What do you think is most important for researchers to understand about your experience since the assault?'

## Analyses

Study results will be shared with the scientific community via conferences and peer-reviewed journal articles. Planned analyses fall into four broad areas.

### Assess the incidence over time of a range of APNS, using both traditional and experimental/state-of-the-art classification systems

No large-scale emergency care-based multisite studies of sexual assault survivors have been conducted, thus the incidence of the full spectrum of APNS over time after sexual assault remains poorly understood. This study will

assess a range of APNS, as well as multivariate patterns of comorbidity among these outcomes and the impact of these outcomes on general physical and mental health over time. APNS will be assessed using both traditional classifications (eg, PTSD, depression, pain and postconcussion symptoms), and also more discrete, homogenous outcomes (eg, avoidance, re-experiencing, numbing, hopelessness, etc). These homogenous outcomes will, to the extent possible, build on the National Institute of Mental Health Research Domain Criteria (RDoC) classification system (https://bit.ly/2pudCZH). Subsequent analyses will attempt to group sexual assault survivors according to the combinations of these discrete outcomes as they develop. Classification systems such as RDoC attempt to overcome two important limitations to traditional classifications: (1) they are not indexed to specific biological processes or components of brain functioning, which often evolved based on the traditional bailiwicks ofspecific medical specialties, (2) each traditional classification captures onlya fragment of a trauma survivor's experience, providing no way to summarize thecomplex patterns of overlapping/co-occurring symptoms across multipletraditional classifications that trauma survivors typically experience.

### Gain new understanding of the pathogenesis of adverse post-traumatic neuropsychiatric outcomes after sexual assault

As noted above, a major barrier to developing more effective preventive interventions is that the pathogenesis of APNS after sexual assault remains poorly understood. An evaluation of risk factors and developmental processes involved in the temporal unfolding of adverse post-traumatic neuropsychiatric outcomes after sexual assault will be evaluated using both traditional and molecular epidemiological methods, with complementary cell culture and animal mechanistic studies where valuable. Available data indicate that biologic samples collected in the immediate aftermath of trauma can provide valuable insights into the pathogenesis of APNS.[6 39–48] The goal of studies performed with these data (eg, genetic, epigenetic, gene expression, miRNA studies) will be to gain improved understanding to inform the development of more effective preventive/recovery interventions for sexual assault survivors.

### Derive risk prediction tools that identify sexual assault survivors at high risk of specific adverse post-traumatic outcome(s)

For more than 50 years, women sexual assault survivors presenting for emergency care after assault have been risk stratified for pregnancy and sexually transmitted disease, and those at high risk have received preventive interventions. However, no such risk stratification tools exist to help identify women at high risk of APNS.[4] Survivor assault characteristics, demographic characteristics, life history, preassault health status, acute post-traumatic symptoms and biological characteristics will be used in an effort to develop clinical decision support tools that identify survivors at high risk of one or more APNS. These clinical

decision support tools will be developed using machine learning (ML) methods,[49] training and validation samples, and internal cross-validation of training samples to minimise over-fitting. A number of different ML algorithms will be examined, such as naïve Bayes, ensemble regression trees (random forests), penalised regression (elastic net), algorithms that allow for non-linearities and non-additivities in stepwise trees with embedded splines (gradient boosting), and support vector machines. The value of combining results across these different algorithms using the super learner ensemble method will also be assessed.[50–54] Tiering and targeting will be used to try to limit the number of items needed in prediction tools. Tiering refers to nested ML analyses based on successively adding more costly predictors to the models, where cost is defined in terms of both time (eg, number of questions that SANE needs to ask to complete) and processing (eg, costs of genetic testing, neuroimaging, etc). Targeting refers to determining subsets of patients that vary in the extent to which prediction accuracy over a clinical decision threshold varies depending on a given level of tiering. For example, screening tests are often used to determine whether individual patients need more complex and expensive tests. The equivalent in our context will be to determine the cross-validated predicted values based on initial predictors that indicate the need for further data.

### Obtain feedback from sexual assault survivors regarding their experiences with healthcare providers, law enforcement and the legal system, and their experiences with the present research study

The lack of large-scale research networks for sexual assault survivors has hampered the ability to obtain feedback from survivors regarding services they receive. Such feedback is critical for continued quality improvement and to ensure that health, law enforcement and legal systems designed for sexual assault survivors are indeed 'survivor-centred'. Similarly, it is critical for research studies to ensure that their protocols are not viewed as negative experiences by sexual assault survivor cohorts taking part in them. Analyses will be performed which evaluate survivor experiences with these services and with the research protocol. Of note, for all analyses in which substantial missing data is present, sensitivity analyses will be performed evaluating analyses results with and without data imputation for missing values.

## ETHICS AND DISSEMINATION

Although the safety of participants is of paramount importance to any research study, this is a matter of particular concern in studies of traumatised individuals. Study participants are told that they may choose not to answer any questions that cause discomfort and that they will be paid the full financial incentive regardless of whether or not they decide to skip self-report questionnaire items or stop the questionnaire. As noted above, study participants who choose to participate at the time

of the emergency care provide written or electronic informed consent only for the initial evaluation. At the beginning of the 1 week assessment, in-person written consent is obtained to perform the 1 week assessment and follow-up assessments. The research assessments never include details of/questions regarding the assault experience itself. This information is obtained from emergency care records. During enrolment in the study, participants are informed that they will be contacted to learn more about the recovery process after trauma. The study is framed in terms of 'recovery', as we believe that it is very important to help create an expectation of healing in study participants since previous research suggests that participant expectation may potentially influence long-term outcomes after trauma.[55] One-week assessments are performed via telephone or self-report computer-based questionnaire. Subsequent surveys at 6 weeks, 6 months and 1 year are also offered either via telephone or self-report computer-based questionnaire. This increases privacy and anonymity and encourages candid participant feedback regarding study participation. Assessment modalities (ie, in-person, online, telephone) are offered depending on participants' preference. This methodology balances our goals of providing as much anonymity as possible while allowing those participants the opportunity to continue to with the study who have travel issues or do not have internet access at a follow-up time point.

Study participants receive the exact same survivor services and follow-up referrals/care as other survivors presenting for care. All survivors are actively referred for mental and physical health services through usual postassault care follow-up services managed by their emergency care providers and are encouraged to utilise a rape crisis advocate to help them receive this care. Participants who report substantial distress and/or who appear to be developing significant anxiety and/or depressive symptoms are encouraged to utilise local mental health services. In addition, study participants are provided with a book (free of charge), *Recovering from Rape*[56] or *Life, Reinvented*,[57] which is written specifically to help women in the early aftermath of sexual assault.

## Concluding summary

The WHS is a unique large-scale emergency care-based longitudinal cohort study of recovery in the 12 months after sexual assault. The study seeks to increase understanding of the incidence, pathogenesis and trajectories of adverse neuropsychiatric sequelae, to develop clinical decision support tools that effectively predict such outcomes, and to obtain feedback from survivors regarding the services they receive. Study success is based on our study participants, who share their experiences with the understanding that researchers will use this information to try to improve the care of survivors in the future. We are grateful for their efforts and recognise our responsibility.

**Author affiliations**
[1]Psychiatry, Medical University of South Carolina, Charleston, South Carolina, USA
[2]Department of Anesthesiology, UNC-Chapel Hill, Chapel Hill, North Carolina, USA
[3]Department of Psychology and Neuroscience and Department of Sociology, University of North Carolina at Chapel Hill, Chapel Hill, North Carolina, USA
[4]Department of Psychiatry, Texas A&M University System Health Science Center College of Medicine, Bryan, Texas, USA
[5]Department of Psychiatry and Behavioral Sciences, Emory University, Atlanta, Georgia, USA
[6]Tulsa Forensic Nursing, Tulsa Police Department, Tulsa, Oklahoma, USA
[7]Forensic Nursing, Cone Health, Greensboro, North Carolina, USA
[8]Forensic Nurse Examining Team, University of Colorado Health Colorado Springs, Colorado Springs, Colorado, USA
[9]Department of Health Care Policy, Harvard Medical School, Boston, Massachusetts, USA
[10]Department of Emergency Medicine, University of North Carolina at Chapel Hill, Chapel Hill, NC, United States

**Contributors** SM conceived of the study. NS, JS, AS, KB, IL, SLM, SAMR, KB, CR, ML, RCK, CN, KW and SM helped to design the final study protocol. All authors reviewed the manuscript.

**Funding** R01AR064700 funded by the following National Institutes of Health Institutes: NIAMS, NINDS, OD (ORWH), NINR, NIMH, and NICHD. This work was supported by NIH R01 AR064700. The content is solely the responsibility of the authors and does not necessarily represent the official views of the NIH.

**Competing interests** In the past 3 years, RCK received support for his epidemiological studies from Sanofi Aventis; was a consultant for Johnson & Johnson Wellness and Prevention, Sage Pharmaceuticals, Shire, Takeda; and served on an advisory board for the Johnson & Johnson Services Inc. Lake Nona Life Project. RCK is a co-owner of DataStat, Inc., a market research firm that carries out healthcare research.

**Patient consent for publication** Not required.

**Ethics approval** All procedures were approved by the UNC IRB (13–3193).

**Provenance and peer review** Not commissioned; externally peer reviewed.

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
