## [Reviewer comments · BMJ Open]

ARTICLE DETAILS

TITLE (PROVISIONAL)	Protocol for the first large-scale emergency care-based longitudinal cohort study of recovery after sexual assault: the Women's Health Study
AUTHORS	Short, Nicole; Sullivan, Jenyth; Soward, April; Bollen, Kenneth; Liberzon, Israel; Martin, Sandra; Rauch, S; Bell, Kathy; Rossi, Catherine; Lechner, Megan; Novak, Carissa; Witkemper, Kristen; Kessler, Ronald; McLean, Samuel

VERSION 1 – REVIEW

REVIEWER	Hyoung Yoon Chang, MD, PhD, MPH Ajou University School of Medicine, Department of Psychiatry South Korea
REVIEW RETURNED	13-Jun-2019

GENERAL COMMENTS	This study presents the protocol of a large-scale prospective study of experiences and recovery among women presenting for emergency care after sexual assault. It is a very important research question that haven't been given enough attention due to stigma and under-report. The design and the writing are good enough to be published, but as a researcher and an clinician in the similar field, I have a few questions and recommendations. 1. Confidentiality is an important issue regarding the data of sexually assaulted survivors. Making inferences from the manuscript, I believe the data are stored in the BTN data coordinating center. I would appreciate if you elaborate more on the security system of the center and additional means that that authors have in place to secure the confidentiality of the data.2. From the manuscript, I understood that the assessments can take in-person or online or via telephone interview. In later analyses, are the authors planning to use the answers of these three types of interviews as one, or are there any other plans to check the consistency between the different types of interview?3. Analyses #4 (Obtaining feedback about the law enforcement and the legal system) is also a very important research question that I have a lot of interest in. However, the description of the assessment is very brief and it was hard to see whether it included any specific content (such as the result of the trial etc.)4. Although the authors did their best to minimize the attrition rate, sexual assault survivors are highly likely to drop off during the follow-up. When planning for statistical analyses, the authors should consider high attrition rate.
--

REVIEWER	Brigid McCaw healthcare consultant, USA
REVIEW RETURNED	21-Jul-2019

GENERAL COMMENTS	I think this is an important study and commend the authors. I think the inclusion of feedback from survivors on their experiences with health care, law enforcement and legal system as well as their experiences with the research study will be an informative addition. I have one primary concern related to the exclusion criteria, see details under Q 3 I have included some comments related to Q 3, 7 I have indicated N/A to Q 7 as I do not have adequate expertise to comment in detail about the ML methods. I have indicated N/A for Q11, 12 as this is a Study protocol, not a manuscript. Is there a missing word(s) on page 8 line 43? “However, such risk stratification tools exist to help identify women at high risk of APNS” ? missing “no”? 3. Is the study design appropriate to answer the research question? In general “yes” but I am concerned about one aspect of the patient inclusion criteria- the exclusion of “living with one’s assailant”. There is significant overlap between sexual assault and intimate partner violence (IPV). Statistics suggest that the majority of women know the person who assaults them and some of those assailants are intimate partners. Women who are raped by an intimate partner may have a different incidence and recovery trajectory of APNS. The exclusion of those women who are “living with their assailant” might exclude abusive spouses and live-in partners, but would not exclude abusive dating relationships, ex-spouses, or partners who do not live together. Therefore the study population will be a mixture of ‘stranger assault’ and sexual assault in the context of IPV. I suggest instead of an exclusion, that additional information be collected about whether the assailant was known to the person, and if the study participant had an intimate relationship (dating, partner, ex-spouse). This would allow inclusion of this information for future analysis. If the Lifetime events checklist does not include prior experience (adult and childhood) of sexual abuse/trauma that should be included. The ACE survey does not include inquiry about sexual abuse/assault after age 18. Research data suggests that hx (childhood and/or adult) may be important in the incidence and trajectory of APNS and/or recovery. I also suggest Trait Resilience be included in the 12 month assessment. A final suggestion: Provide a free text opportunity for study participants to offer their thoughts/reflections about whether the survey questions (Pain, depression, anxiety, PCL-5, somatic symptoms) included elements that they feel are important in their recovery process. This would be in addition to the RRPQ which focuses on participating in research. In a 6 wk, 6 mo, and 1 yr followup of IPV survivors, we found that the qualitative information from free text questions was more informative than our standard survey questions about mood, stress, and safety and suggested some elements of recovery we hadn’t included. For example- “empowerment” “feeling less isolated” “improved self esteem” “ready to reach out to others” “less shame” were written in ‘free text’ sections and would not have been captured.
--

	7. If statistics are used are they appropriate and described fully? I recommend having manuscript reviewers with expertise in ML methods and algorithms in addition to those with content expertise in sexual assault and mental health. 11. Are the discussion and conclusions justified by the results? Not applicable as this is a proposed study protocol. 12. Are the study limitations discussed adequately? Not applicable as this is a proposed study protocol.
--	--

VERSION 1 – AUTHOR RESPONSE

Reviewer 1:

1. Confidentiality is an important issue regarding the data of sexually assaulted survivors. Making inferences from the manuscript, I believe the data are stored in the BTN data coordinating center. I would appreciate if you elaborate more on the security system of the center and additional means that that authors have in place to secure the confidentiality of the data. Thank you for this question. We have added a section entitled “Confidentiality and Security of Participant Data” (page 5) to provide further details.

2. From the manuscript, I understood that the assessments can take in-person or online or via telephone interview. In later analyses, are the authors planning to use the answers of these three types of interviews as one, or are there any other plans to check the consistency between the different types of interview? Prior research demonstrates measurement invariance for well-validated self-report measures between pen and paper and online/telephone assessments (Cole, Bedeian, & Field, 2006; Rosenbaum, Rabenhorst, Reddy, Fleming, & Howells, 2006; Weigold, Weigold, & Russell, 2013). However, we do track how the data were collected (e.g., phone vs. internet vs. in-person), so it will be possible to test whether the measurement types were equivalent prior to analyzing the data.

3. Analyses #4 (Obtaining feedback about the law enforcement and the legal system) is also a very important research question that I have a lot of interest in. However, the description of the assessment is very brief and it was hard to see whether it included any specific content (such as the result of the trial etc.) Thank you for this question. We have now added additional information regarding legal information collected (see Page 7).

4. Although the authors did their best to minimize the attrition rate, sexual assault survivors are highly likely to drop off during the follow-up. When planning for statistical analyses, the authors should consider high attrition rate. Thank you for this excellent point. We have added information in this regard to the manuscript (page 8): Of note, for all analyses in which substantial missing data is present, sensitivity analyses will be performed evaluating analyses results with and without data imputation for missing values.

Reviewer 2:

1. Is there is a missing word(s) on page 8 line 43? “However, such risk stratification tools exist to help identify women at high risk of APNS” ? missing “no”? Yes, we have edited to correct this and add the missing “no.”

2. Is the study design appropriate to answer the research question? In general “yes” but I am concerned about one aspect of the patient inclusion criteria- the exclusion of “living with one’s

assailant". There is significant overlap between sexual assault and intimate partner violence (IPV). Statistics suggest that the majority of women know the person who assaults them and some of those assailants are intimate partners. Women who are raped by an intimate partner may have a different incidence and recovery trajectory of APNS. The exclusion of those women who are "living with their assailant" might exclude abusive spouses and live-in partners, but would not exclude abusive dating relationships, ex-spouses, or partners who do not live together. Therefore the study population will be a mixture of 'stranger assault" and sexual assault in the context of IPV. I suggest instead of an exclusion, that additional information be collected about whether the assailant was known to the person, and if the study participant had an intimate relationship (dating, partner, ex-spouse). This would allow inclusion of this information for future analysis.

Thank you for this important point. We agree that women who are not living with an assailant may also have ongoing traumatization. We also agree that no boundary, including the one funded/approved for this study by NIH during several rounds of NIH submissions and reviewer feedback (not living with the assailant), will separate out ongoing interactions with or influences by the assailant. We collect data on the nature of the relationship between the survivor and assailant, including whether they are relatives, current or former romantic partners, friend/acquaintance, planned first encounter, stranger, or patient does not remember. This information has now been added to the manuscript (page 6).

3. If the Lifetime Events Checklist does not include prior experience (adult and childhood) of sexual abuse/trauma that should be included. The ACE survey does not include inquiry about sexual abuse/assault after age 18. Research data suggests that hx (childhood and/or adult) may be important in the incidence and trajectory of APNS and/or recovery.

We agree that this is critical data to collect, and we appreciate that our initial manuscript submission was not clear on this point. We have now clarified in the manuscript (page 6) that the LEC does inquire as to whether the participants experienced sexual assault in their lifetime, while the ACE assesses childhood sexual assault. These data will allow evaluation of whether women experienced sexual assault prior to the recent sexual assault, and whether this occurred during childhood or adulthood.

4. I also suggest Trait Resilience be included in the 12 month assessment.

We include trait resilience as a potential predictor of APNS and recovery following trauma. It is a trait characteristic, and should hopefully change relatively less, and in final NIH-approved protocol this was only assessed once. We agree that in future research it may be helpful to assess whether resilience changes after sexual assault.

5. A final suggestion: Provide a free text opportunity for study participants to offer their thoughts/reflections about whether the survey questions (Pain, depression, anxiety, PCL-5, somatic symptoms) included elements that they feel are important in their recovery process. This would be in addition to the RRPQ which focuses on participating in research. In a 6 wk, 6 mo, and 1 yr followup of IPV survivors, we found that the qualitative information from free text questions was more informative than our standard survey questions about mood, stress, and safety and suggested some elements of recovery we hadn't included. For example- "empowerment" "feeling less isolated" "improved self esteem" "ready to reach out to others" "less shame" were written in 'free text' sections and would not have been captured.

Thank you for this comment. At each follow-up time point (1-week, 6-week, 6-month, and 1-year) the questionnaire includes the following question: "What do you think is most important for researchers to understand about your experience since the assault?" This has been clarified in the manuscript (Page 7).

We are very grateful to the reviewers for their time and effort. We hope and believe that their comments have allowed us to improve the manuscript.

References

Cole, M. S., Bedeian, A. G., & Feild, H. S. (2006). The measurement equivalence of web-based and paper-and-pencil measures of transformational leadership: A multinational test. *Organizational Research Methods*, 9(3), 339-368.

Rosenbaum, A., Rabenhorst, M. M., Reddy, M. K., Fleming, M. T., & Howells, N. L. (2006). A comparison of methods for collecting self-report data on sensitive topics. *Violence and Victims*, 21(4), 461.

Weigold, A., Weigold, I. K., & Russell, E. J. (2013). Examination of the equivalence of self-report survey-based paper-and-pencil and internet data collection methods. *Psychological Methods*, 18(1), 53.

VERSION 2 – REVIEW

REVIEWER	Hyoung Yoon CHANG Ajou University School of Medicine, South Korea
REVIEW RETURNED	23-Sep-2019

GENERAL COMMENTS	The authors have answered well to my inquiries and comments, and I believe the manuscript is ready to be published.
---

REVIEWER	Brigid McCaw Retired, Kaiser Permanente USA
REVIEW RETURNED	09-Sep-2019

GENERAL COMMENTS	All concerns were well addressed. Excellent and important study.
--